A late surviving Pliocene seal from high latitudes of the North Atlantic realm: the latest monachine seal on the southern margin of the North Sea

Dewaele Leonard leonard.dewaele@ugent.be 1 2
Lambert Olivier 1
Louwye Stephen 2
1 Earth and History of Life, Royal Belgian Institute of Natural Sciences , Brussels , Belgium
2 Department of Geology, Ghent University , Ghent , Belgium
Pyenson Nicholas
Electronic publication date: 2018 Oct 9
Publication date: 2018
Volume: 6
Electronic Location ID: e5734
Received 2018 Feb 15; Accepted 2018 Sep 5
Copyright: ©2018 Dewaele et al.
Copyright year: 2018
Copyright holder: Dewaele et al.
License: This is an open access article distributed under the terms of the Creative Commons Attribution License, which permits unrestricted use, distribution, reproduction and adaptation in any medium and for any purpose provided that it is properly attributed. For attribution, the original author(s), title, publication source (PeerJ) and either DOI or URL of the article must be cited.
License URL: https://creativecommons.org/licenses/by/4.0/

Keywords: Mammalia, Phocidae, Monachinae, Pliocene, North Atlantic, North Sea

Funding: Research Foundation—Flanders (FWO) 11V9117N Leonard Dewaele was financially supported by the Research Foundation—Flanders (FWO) through an FWO PhD Fellowship (No. 11V9117N). The funders had no role in study design, data collection and analysis, decision to publish, or preparation of the manuscript.

==============================
Background

The family of true seals, the Phocidae, is subdivided into two subfamilies: the southern Monachinae, and the northern Phocinae, following the subfamilies’ current distribution: extant Monachinae are largely restricted to the (sub-)Antarctic and the eastern Pacific, with historical distributions of the monk seals of the genus Monachus in the Caribbean, the Mediterranean and around Hawaii; and Phocinae to the northern temperate and Arctic zones. However, the fossil record shows that Monachinae were common in the North Atlantic realm during the late Miocene and early Pliocene. Until now, only one late Pliocene record is known from the Mediterranean, Pliophoca etrusca from Tuscany, Italy, but none from farther north in the North Atlantic.

Methods

We present the description of one partial phocid humerus collected in the early 20th century from the Antwerp area (Belgium), with an assessment of its stratigraphic origin using data from the literature.

Results

The studied humerus was recovered during construction works at the former Lefèvre dock in the Antwerp harbour (currently part of the America dock). Combining the information associated to the specimen with data from the literature and from local boreholes, the upper Pliocene Lillo Formation is ascertained as the lithological unit from which the specimen originates. Morphologically, among other features the shape of the deltopectoral crest and the poor development of the supinator crest indicates a monachine attribution for this specimen. The development of the deltopectoral crest is closer to the condition in extant Monachinae than in extinct Monachinae.

Discussion

The presented specimen most likely represents a monachine seal and a literature study clearly shows that it came from the latest early to late Pliocene Lillo Formation. This would be the first known monachine specimen from the latest early to late Pliocene of the North Sea, and more broadly from the northern part of the North Atlantic realm. This humerus differs from the humerus of P. etrusca and suggests a higher diversity of Monachinae in the latest early to late Pliocene than previously assumed.

Introduction

True seals (Mammalia, Pinnipedia, Phocidae) are subdivided into two extant subfamilies: Monachinae Gray, 1869 and Phocinae Gray, 1821. Some researchers also accept the existence of a third extant subfamily: Cystophorinae Gray, 1866, including the hooded seal Cystophora cristata (Erxleben, 1777) and the elephant seals Mirounga angustirostris Gill, 1866, and Mirounga leonina (Linnaeus, 1758) (e.g., Scheffer, 1958; King, 1964; Chapskii, 1974; Koretsky & Rahmat, 2013). However, this is based on few morphological features, such as the dental formula (I2/1) and the presence of a proboscis (e.g. King, 1964), while more extensive morphological and molecular phylogenetic analyses do not support the identification of this third extant subfamily (e.g. Berta & Wyss, 1994; Bininda-Emonds & Russell, 1996; Árnason et al., 2006; Higdon et al., 2007; Fulton & Strobeck, 2010). Koretsky & Holec (2002) erected a fourth subfamily of Phocidae, only represented by the extinct genus Devinophoca Koretsky & Holec, 2002. However, a recent phylogenetic analysis by Dewaele et al. (2017) suggested that the genus Devinophoca may represent a stem Phocinae. Both subfamilies of Monachinae and Phocinae are characterized by different biogeographic ranges for the extant taxa (King, 1964). Following their current biogeographic distribution, Monachinae can be considered to be southern phocids, while Phocinae can be considered boreal phocids. Indeed, the geographic range of Phocinae is restricted to the Arctic and Northern temperate zones, including the Caspian Sea (Pusa caspica (Gmelin, 1788)) and Lake Baikal (Pusa sibirica (Gmelin, 1788)), while most Monachinae live more southerly (e.g., King, 1964; Jefferson, Webber & Pitman, 2008; Würsig, Thewissen & Kovacs, 2017). The Lobodontini Gray, 1869 tribe lives in the Antarctic and sub-Antarctic waters. The elephant seals of the genus Mirounga Gray, 1827 live in sub-Antarctic waters, along the western shores of South America, in the Southwest Atlantic, at Puerto Madryn in Argentina, but also in the Northeast Pacific, from California to Alaska. The monk seals (genera Monachus Fleming, 1822 and Neomonachus Scheel, Slater, Kolokotronis, Potter, Rotstein, Tsangaras, Greenwood & Helgen, 2014) have a sub-tropical to tropical distribution, restricted to the Mediterranean (Monachus monachus Hermann, 1779), the Caribbean Sea (Neomonachus tropicalis (Gray, 1850), recently extinct) and the Hawaiian Islands, in the central Pacific Ocean (Neomonachus schauinslandi (Matschie, 1905)) (e.g., King, 1964; Jefferson, Webber & Pitman, 2008; Würsig, Thewissen & Kovacs, 2017).

However, the distribution of extant Monachinae does not reflect the past distribution of Monachinae and Phocinae. Indeed, during the Neogene, multiple monachine taxa lived in the North Atlantic realm, with fossils of Auroraphoca atlantica Dewaele, Peredo, Meyvisch & Louwye, 2018, Callophoca obscura Van Beneden, 1876, and Virginiaphoca magurai Dewaele, Peredo, Meyvisch & Louwye, 2018, from late Miocene deposits from Belgium and late Miocene and early Pliocene deposits from the east coast of North America (Van Beneden, 1876; Van Beneden, 1877; Ray, 1976; Koretsky & Ray, 2008; Dewaele, Lambert & Louwye, 2018). Historically, the youngest published fossil monachine taxon of the Northern Hemisphere is the holotype of Pliophoca etrusca Tavani, 1941 from the Piacenzian (late Pliocene) of Tuscany, Italy (Tavani, 1941; Berta et al., 2015).

Materials and Method

Biological sample

This study focuses on specimen IRSNB M2308. Fossil comparison material includes all known late Miocene-early Pliocene Monachinae from the North Atlantic realm: Auroraphoca atlantica, Callophoca obscura, Homiphoca (capensis), P. etrusca, and Virginiaphoca magurai, as well as other Neogene Monachinae from the Southern Hemisphere: Acrophoca longirostris De Muizon, 1981, Australophoca changorum Valenzuela-Toro, Pyenson, Gutstein & Suárez, 2015, Piscophoca pacifica De Muizon, 1981, and Properiptychus argentinus (Ameghino, 1893), based on personal observations and information retrieved from the literature. The sample of comparison material also includes representatives of nearly all extant monachine genera, housed at the IRSNB: leopard seal Hydrurga leptonyx (Blainville, 1820), Weddell seal Leptonychotes weddellii (Lesson, 1826), crabeater seal Lobodon carcinophaga (Hombron & Jacquinot, 1842), Ross seal Ommatophoca rossii (Gray, 1844), southern elephant seal Mirounga leonina, and Mediterranean monk seal Monachus monachus. Extant and extinct Phocinae are considered from a more general perspective. Dewaele, Lambert & Louwye (2018) renamed Monotherium aberratum Van Beneden, 1876 and Monotherium affine to Frisiphoca aberratum and Frisiphoca affine, respectively. In the current study, the specific names are corrected to Frisiphoca aberrata and Frisiphoca affinis to be grammatically in order. It should also be noted that the phocine affinities of the genus Frisiphoca are based on few and relatively weak characters and that the genus may as well be monachine (see Dewaele, Lambert & Louwye, 2018). Specimens of extant Phocinae considered for this study include specimens housed at the IRSNB and USNM. Extinct Phocinae include specimens housed at the IRSNB, MNHN, and USNM, as well as specimens published in the literature.

Historical and geological context of humerus IRSNB M2308

Humerus IRSNB M2308 was discovered in 1904 by the private collector Georges Hasse. The collection of the latter entered into the RBINS collection in the 1910s. The data provided by the labels adjoining specimen IRSNB M2308 state only “Anvers” (Antwerp) and “bassin-canal” as the locality of the specimen (Fig. 1). The specimen was originally considered to represent a humerus of Prophoca Van Beneden, 1876.

Figure 1 Labels found associated to the humerus IRSNB M2308, Monachinae indet.

(A), original label, stating Antwerp (“Anvers”) as the origin of the specimen and 1904 as the year of discovery. A provisional, unpublished and unsupported identification returned Prophoca Van Beneden, 1876; (B), more recent label, stating the more precise locality as one of the docks in the Antwerp harbour area (“bassin-canal”).

Originally, the specimen has been stratigraphically assigned to the “Poederlian” (Poederlien, Fig. 1A). However, the Poederlian is currently a disused regional Neogene stage (Laga & Louwye, 2006). Laga & Louwye (2006) argue that the stage has never been defined properly, and that different historic authors employed different interpretations of the stage, and that the type locality is unsuitable for a stage type section. The Poederlian is named after the Belgian village of Poederlee, roughly 30 km east of Antwerp, and the so-called Poederlian deposits in the Antwerp harbour area were correlated to the deposits at Poederlee by Vincent (1889). Later authors disagreed with Vincent (1889), and considered the Poederlian in the Antwerp harbour area to represent the upper substage of the Scaldisian (Leriche, 1922).

Fortunately, Hasse (1909) described Poederlian walruses from the Antwerp harbour area, providing geographic maps, photographs of stratigraphic sections, and a detailed description of the lithology. Hasse (1909) states that these walrus specimens were discovered during construction works for new docks (“basin” in French) in 1902–5, alongside other fossil mammal remains including phocid remains (presented as Phoca). The time interval, location data, and stratigraphic data from Hasse (1909) match perfectly the labels of IRSNB M2308 (Fig. 1), and it can be safely assumed that specimen IRSNB M2308 had been found at the same locality, and in the same levels (Poederlian), as the walruses that he described (now attributed to Ontocetus emmonsi Leidy, 1859, see Kohno & Ray, 2008). Hasse (1909) pinpointed the geographic setting to the Lefèvre dock (Bassin Lefèvre). Currently, the Lefèvre dock is merged into the America dock, forming its southeastern portion (Fig. 2). Additionally, Hasse (1909) presented malacological data for the Lefèvre dock fossil-bearing level; one of the most common taxa is the gastropod Fusus contrarius (Linnaeus, 1771). More recent research renamed the fossil F. contrarius to Neptunea angulata (Wood, 1848) to make the distinction with extant F. contrarius. In Neogene deposits of the Antwerp area, N. angulate is considered a characteristic taxon for the Oorderen Sands and the overlying Kruisschans Sands members of the Pliocene Lillo Formation (Fig. 3A) (Nyst, 1843; Marquet, 1993; Marquet, 1997; Marquet, 1998). The Oorderen Sands Member overlies the Luchtbal Sands Member, the lowest member of the Lillo Formation, conformably. Another mollusc from the locality and level listed by Hasse (1909) is the bivalve Cardium parkinsoni, which Tavernier & Heinzelin (1962) restricted to the Kruisschans Sands and Merksem Sands members (Fig. 3A).

Figure 2 Geological map.

(A), Regional map of the southern part of the North Sea Basin, with bordering countries. Capital cities labelled in yellow, and the Antwerp area labelled in red. (B), Cenozoic geological map of the Antwerp area, showing the location of specimen IRSNB M2308 in the Antwerp harbour area. (C), Stratigraphic legend for the Paleogene and Neogene strata from the Antwerp area, based on data from Dienst Ondergrond Vlaanderen (DOV; http://dov.vlaanderen.be). Abbreviations: NL, Netherlands; GER, Germany; LUX, Luxemburg; FRA, France; UK, United Kingdom; BEL, Belgium; Lux., Luxemburg City; Plei., Pleistocene; Plio., Pliocene; Pi., Piacenzian; Za., Zanclean; Mes., Messinian; Ser., Serravallian; Lang., Langhian; Burdigal., Burdigalian; Aq., Aquitanian; Priabon., Priabonian; Barton., Bartonian; Fm., Formation. (Adapted from Dewaele, Lambert & Louwye, 2017).

Figure 3 Pliocene stratigraphy of the Antwerp harbour region.

(A), stratigraphic column showing the succession of the different members of the uppermost lower and upper Pliocene Lillo Formation in the Antwerp harbour area. (B), simplified lithological of the section from Lefèvre Dock where Hasse (1909) discovered specimen IRSNB M2308. Litholog drawn after descriptions by Hasse (1909).

Borehole logs (GEO-04/169-BRO-B1 and kb15d28w-B211; Dienst Ondergrond Vlaanderen, http://www.dov.vlaanderen.be) within close proximity of the locality analysed by Hasse have shown that the lower—upper Pliocene Lillo Formation is underlain by the lower Pliocene Kattendijk Formation in the area. However, the boundary between the Lillo and Kattendijk formations is consistently located at ten meters or more below the core top, while Hasse (1909) clearly stated that the walrus fossils (and associated phocid material) he found came from less than three meters below the top of the section (Fig. 3B). Consequently, all arguments confirm the Lillo Formation as the origin of both the walruses described by Hasse (1909) (for more details see Kohno & Ray, 2008) and the phocid humerus IRSNB M2308.

Dinoflagellate cyst biostratigraphy by De Schepper, Head & Louwye (2009) dated the Oorderen Sands Member no younger than 2.72–2.74 Ma, but not older than the maximum possible age of 3.71 Ma for the Lillo Formation, and the upper boundary of the Kruisschans Sands Member to be no younger than 2.58 Ma. These two members are thus included in an interval ranging from the latest Zanclean (latest early Pliocene) to the Piacenzian (late Pliocene).

Systematic paleontology

Pinnipedia Illiger, 1811	
Phocidae Gray, 1821	
Monachinae Gray, 1869	
Indeterminate Monachinae	

Referred Specimen—IRSNB M2308, right humerus, Oorderen Sands or Kruisschans Sands members, Lillo Formation, America dock, Antwerp, Belgium.

Locality—Historically “Anvers (bassin-canal),” but currently reconsidered as the southeastern area of the America dock in the Antwerp Harbour area, north to northwest of the city of Antwerp, Antwerp province, Belgium, following data from Hasse (1909) (see discussion above).

Stratigraphy and Age—Historically “Poederlien,” but currently reconsidered to belong to either the Oorderen Sands or the Kruisschans Sands members of the Lillo Formation, following data from Hasse (1909) and De Schepper, Head & Louwye (2009), between 2.58 Ma and 3.71 Ma (see discussion above). This entails most likely a Piacenzian age (late Pliocene), although a latest Zanclean (latest early Pliocene) age cannot be completely ruled out.

Description and Comparison—Specimen IRSNB M2308 was found isolated, and no other phocid remains are currently known from the late Pliocene Lillo Formation of Antwerp, Belgium. IRSNB M2308 is a partial right humerus, lacking the distal epiphysis. The distal portion of the diaphysis is fractured, with the internal bone structure clearly visible. Consequently, it is clear that the distal part is not missing due to skeletal immaturity and non-fusion of the distal epiphysis.

The preserved portion of humerus IRSNB M2308 is 123.2 mm long, allowing us to assume that the length of the complete humerus should have been at least 140–150 mm, and that the individual must have been comparable in size to the extinct monachines Homiphoca sp. from the early Pliocene of South Africa and Piscophoca pacifica from the late Miocene to early Pliocene of Peru (see Muizon & Hendey, 1980; De Muizon, 1981), and the extant monachine Leptonychotes weddellii (2.7–3.3 m total length; from King, 1964). However, this is still considerably smaller than the humerus of the monachine Callophoca obscura and the phocine Platyphoca vulgaris Van Beneden, 1876, from the early Pliocene Kattendijk Formation, underlying the Lillo Formation in the Antwerp harbour area, and the early Pliocene Yorktown Formation in the Lee Creek Mine, Aurora, North Carolina (Figs. 4A–4D, versus 4E). In addition, IRSNB M2308 is much larger than the humerus of the monachine Properiptychus argentinus from the middle Miocene of Argentina, and shorter than the holotype humeri of Acrophoca longirostris from the late Miocene to early Pliocene of Peru, Auroraphoca atlantica from the early Pliocene of the USA, and the presumed phocine Frisiphoca affinis (Van Beneden, 1876) from the late Miocene of Belgium (De Muizon, 1981; Muizon & Bond, 1982; Dewaele et al., 2018). However, it is longer than the humeri of the other fossil Phocinae from the Neogene of the North Sea Basin (see Van Beneden, 1877; (Koretsky, 2001; Koretsky & Peters, 2008; Koretsky, Rahmat & Peters, 2014; Koretsky, Peters & Rahmat, 2015) and the humeri of many other fossil Monachinae from the North and South Atlantic and the eastern South Pacific (see Muizon & Hendey, 1980; Valenzuela-Toro et al., 2015; Dewaele et al., 2018). Although the illustrated humerus of Callophoca obscura is approximately 150 mm long (Fig. 4E), Koretsky & Ray (2008) identified C. obscura and Mesotaria ambigua Van Beneden, 1876 as being conspecific, noting no morphological differences except for the size. The illustrated humerus of C. obscura represents a smaller, most likely female, specimen of C. obscura.

Figure 4 Humerus IRSNB M2308 and comparison material.

(A–D), right humerus IRSNB M2308, Monachinae indet. (Antwerp, Belgium; late Pliocene), in (A), medial view; (B), anterior view; (C), lateral view; (D), posterior view. (E), left humerus of Callophoca obscura (USNM 186944) (Lee Creek Mine, Aurora, North Carolina, U.S.A.; Zanclean) in medial view (from Dewaele et al., 2018); (F), left humerus of Pliophoca etrusca (MSNUP I-13993, holotype) (Casa Nuova, Tuscany, Italy; Piacenzian) in medial view (image courtesy: G. Bianucci); (G) schematic drawing of left humerus of the extant monachine Monachus monachus; (H), schematic drawing of left humerus of the extant phocine Phoca vitulina (G, H: redrawn from Valenzuela-Toro et al., 2015). Scale bar equals 5 cm.

The humeral head is prominent and hemispherical in IRSNB M2308, with a clear, sharp notch between the head and the neck, similar to the extant Leptonychotes weddellii (Lesson, 1826. It is less prominent and hemispherical in IRSNB M2308 than in the extinct Piscophoca pacifica, but slightly better developed than in other Monachinae. Among Phocinae, the extinct Cryptophoca pontica (Nordmann, 1860) and Leptophoca proxima (Van Beneden, 1876) have a similarly large humeral head in comparison to the rest of the bone (Dewaele, Lambert & Louwye, 2017). A hemispherical humeral head overhanging the diaphysis posteriorly is common among Phocinae and is present in, for instance, the extant bearded seal Erignathus barbatus Erxleben, 1877, gray seal Halichoerus grypus (Fabricius, 1791), ribbon seal Histriophoca fasciata (Zimmermann, 1783), and harp seal Pagophilus groenlandicus (Erxleben, 1777). Contrastingly, the humeral head more strongly overhangs the diaphysis posteriorly in the early Pliocene phocine Phocanella pumila Van Beneden, 1876, and in the contemporaneous monachine P. etrusca than in IRSNB M2308. In P. etrusca, the orientation of the humeral head is more posterior (almost completely posterior), while its orientation is posteroproximal in IRSNB M2308 (compare Figs. 4C and 4F). The humeral head in IRSNB M2308 is slightly compressed anteroproximally (height-to-width ratio is 42.1 mm: 44.6 mm; Table 1). The posterodistal margin of the humeral head is subtriangular in IRSNB M2308. We observed a similar condition in Monachus monachus, while it tends to be more smoothly rounded in Lobodontini and in Phocinae.

Table 1 Measurements of humerus IRSNB M2308 and humeri of other Monachinae from the Pliocene of the North Atlantic realm.

Measurements taken to the nearest 0.1 mm with an analog caliper. Taking measurements follows the approach outlined by Koretsky (2001). Measurements for P. etrusca retrieved from the description by Berta et al. (2015). The measurements for Auroraphoca atlantica have been published by Dewaele et al. (2018).

Measurement	IRSNB M2308	Auroraphoca atlantica USNM 181419	Callophoca obscura USNM 186944	Homiphoca sp. USNM 187228	Pliophoca etrusca MSNUP-I-13993	
Total length	N/A	159.3	150.7	128.0	125	
Height humeral head	42.1	38.9	36.5	N/A	37	
Width humeral head	44.6	44.3	41.2	39.7	32	
Transverse width proximal epiphysis	63.0	62.5	63.1	58.7	62	
Anteroposterior width	79.9	74.0	77.2	69.3	N/A	
Proximodistal length deltopectoral crest	92.2	100.3	95.5	88.0	88	
Transverse width diaphysis	27.4	28.5	27.9	25.6	23	

The lesser does not reach the level of the humeral head, proximally. This condition varies between extant and extinct Phocidae (e.g. De Muizon, 1981; Koretsky, 2001; Dewaele, Lambert & Louwye, 2017; Dewaele, Lambert & Louwye, 2018; Dewaele et al., 2017; Dewaele et al., 2018). Apart from M. monachus, all extant Monachinae have a lesser tubercle that is well-developed, exceeding the proximal level of the humeral head; while in extinct Monachinae, the lesser tubercle usually does not exceed the proximal level of the humeral head, except in Callophoca obscura, Homiphoca sp. (Hendey & Repenning, 1972), Pliophoca etrusca, Properiptychus argentinus and Virginiaphoca magurai (Muizon & Hendey, 1980; De Muizon, 1981; Muizon & Bond, 1982; Berta et al., 2015; Dewaele, Lambert & Louwye, 2018). The greater tubercle on the humerus IRSNB M2308 reaches proximal of the humeral head, whereas the greater tubercle is generally little-developed in extant Monachinae, not exceeding the proximal level of the humeral head (De Muizon, 1981). However, in extinct Monachinae, this condition varies, with the greater tubercle exceeding the proximal level of the humeral head in most taxa (Acrophoca longirostris, Auroraphoca atlantica, C. obscura, Homiphoca sp. (Muizon & Hendey, 1980), and Piscophoca pacifica, and also IRSNB M2308), but not in others (Pliophoca etrusca, and Properiptychus argentinus) (De Muizon, 1981; Muizon & Bond, 1982; Berta et al., 2015; Dewaele, Lambert & Louwye, 2018). Among Phocinae, all extant taxa have a lesser tubercle that exceeds the humeral head proximally. However, as with Monachinae, this condition varies among extinct Phocinae: Batavipusa neerlandica Koretsky & Peters, 2008, Frisiphoca sp., Leptophoca proxima, Phocanella pumila, Praepusa sp., and Sarmatonectes sintsovi Koretsky, 2001 are characterized by a lesser tubercle that does not reach the level of the humeral head proximally (Koretsky, 2001; Koretsky & Peters, 2008; Koretsky & Ray, 2008; Dewaele et al., 2017; Dewaele, Lambert & Louwye, 2017; Dewaele, Lambert & Louwye, 2018); Cryptophoca maeotica and Nanophoca vitulinoides (Van Beneden, 1871) have a lesser tubercle that reaches the level of the humeral head (Koretsky, 2001; Dewaele et al., 2017); and Monachopsis pontica (Eichwald, 1850) has a lesser tubercle that exceeds the level of the humeral head, proximally (Koretsky, 2001). A greater tubercle exceeding the humeral head has been observed in B. neerlandica and Praepusa sp. (Koretsky, 2001; Koretsky & Peters, 2008).

In anteroproximal view, the proximal portion of the deltopectoral crest of IRSNB M2308 is strongly curved medially, yielding a deep and relatively narrow bicipital groove, i.e., that is as deep as it is wide. This condition differs from other Monachinae, having bicipital grooves that are usually wider than deep. This groove is moderately deep in Hydrurga leptonyx (Blainville, 1820) and Leptonychotes weddellii (see De Muizon, 1981). The bicipital groove of IRSNB M2308 is smooth, as in H. leptonyx, and L. weddellii, while other extant and extinct Monachinae have a transverse bar at the proximal portion of the bicipital groove (see De Muizon, 1981; Dewaele, Lambert & Louwye, 2018). Phocinae generally have a rather narrow bicipital groove, narrower than in IRSNB M2308, and they lack a transverse bar in the bicipital groove.

Figure 5 Deltopectoral crest shape variation.

The left column (A–K) shows humeri of different taxa of extinct and extant Monachinae in lateral view. The right column (L–S) shows humeri of different taxa of extinct and extant Phocinae in lateral view. Notwithstanding overlap in age ranges, humeri of geologically older taxa are listed higher in the figure and humeri of geologically younger, i.e., extant, taxa are listed below. Given the incompleteness of humerus IRSNB M2308 (H), quantification of the shape of the deltopectoral crest through measurements is hampered. However, a qualitative comparison with Phocinae and other Monachinae (A–G, I–K) and Phocinae (L–S) shows that the deltopectoral crest is much more curving in lateral view than the diaphysis. For easy comparison, each illustrated specimen is accompanied by the highly stylized outlines of the deltopectoral crest and diaphysis in lateral view. Light gray indicates little difference in curvature between the deltopectoral crest and the diaphysis. Dark gray and black indicate a deltopectoral crest that is slightly more curving than the diaphysis, or much more curving than the diaphysis, respectively. A strongly-curving deltopectoral crest is indicative for Monachinae. Drawings after photographs from Koretsky (2001) (N, O, Q), Berta et al. (2015) (I), Valenzuela-Toro et al. (2015) (B, J, K), Dewaele, Lambert & Louwye (2017), Dewaele, Lambert & Louwye (2018) (L, M), Dewaele et al. (2017), Dewaele et al. (2018), (A, C, F, P), and personal observations (L Dewaele, pers. obs., 2018; D, E, G, H, R, S). Gray areas on bones represent broken or obliterated areas. Some images have been mirrored for consistency.

Overall, the deltopectoral crest of IRSNB M2308 is typically monachine in lateral view, in that the deltopectoral crest curves regularly from the greater tubercle, proximally, and smoothly merges into the diaphysis, distally (e.g. King, 1964; De Muizon, 1981; Berta & Wyss, 1994) (Figs. 4A–4G, versus Fig. 4H). While the deltopectoral crest of extant Phocinae terminates abruptly, distally (Fig. 3H), recent studies of extinct Phocinae suggest that also some extinct Phocinae have a deltopectoral crest that relatively smoothly contacts the diaphysis, distally (e.g., Koretsky, 2001; Dewaele et al., 2017; Dewaele, Lambert & Louwye, 2017). Indeed, the deltopectoral crest is rather rounded in the presumed fossil Phocinae Cryptophoca maeotica, Kawas benegasorum Cozzuol, 2001, and Sarmatonectes sintsovi, and the deltopectoral crest of Leptophoca proxima and Prophoca rousseaui Van Beneden, 1876 terminates close to the distal epiphysis of the humerus (Dewaele, Lambert & Louwye, 2017; and references therein). However, a characteristic that differs between Monachinae and Phocinae, both extant and extinct, is the angular (e.g., Acrophoca longirostris, Piscophoca pacifica) to regularly convex (e.g., Australophoca changorum, Homiphoca sp., and Monachus sp.) subtriangular outline of the deltopectoral crest observed in lateral view in Monachinae. This feature can also be described as a curvature of the anterior margin of the deltopectoral crest in lateral view that is much stronger than the curvature of the posterior portion of the diaphysis (Fig. 5). In extinct Phocinae that appear to have a smoothly curving deltopectoral crest, the degree of curvature of the deltopectoral crest does not differ significantly from the curvature of the diaphysis in general (see Koretsky, 2001; Dewaele, Lambert & Louwye, 2017). This study stresses the need for morphometric analyses better quantifying morphological differences between the humeri of Monachinae and Phocinae. Unfortunately, quantification of the shape of the deltopectoral crest, and the humerus of phocids in general, is outside the scope of the present study. The deltopectoral crest of IRSNB M2308 is roughly angular in lateral view, corresponding thus with Monachinae, rather than with Phocinae. The maximum breadth of the deltopectoral crest, in lateral view, is located at approximately at the proximal 1/3 of the total length of the bone in IRSNB M2308. In lateral view, the humerus IRSNM M2308 has a strongly anteriorly projected deltopectoral crest (at the level of the deltoid tuberosity), as in most Pliocene and extant Monachinae. However, the distal portion of the deltopectoral crest merges more gradually into the distal region (or end) of the diaphysis in IRSNB M2308 than in extant Monachinae (except Monachus spp. and Ommatophoca rossii). Medial on the distally tapering edge of the deltopectoral crest, a rugose area on the deltopectoral crest marks the insertion area of the pectoralis muscle. The insertion area of the pectoralis muscle extends clearly distal to halfway the humerus in IRSNB M2308. The location of the insertion of the pectoralis muscle on the humerus is another difference in the morphology of the deltopectoral crest in Monachinae and Phocinae (De Muizon, 1981). While it reaches distal to halfway the diaphysis, and distal to the trochideltoid surface in Monachinae (e.g., Bryden, 1971; De Muizon, 1981; Muizon & Bond, 1982; C de Muizon, pers. comm., 2018), the insertion of the pectoralis muscle does not appear to extend distal to the trochideltoid surface in (extant) Phocinae (L Dewaele, pers. obs., 2018; C de Muizon, pers. comm., 2018). This characteristic separating Monachinae from Phocinae may also spur the revision of Prophoca rousseaui. A phylogenetic analysis by Dewaele, Lambert & Louwye (2017) returned the species as a phocine seal. Yet, the lectotype humerus has an insertion area of the pectoralis muscle similar to the condition in Monachinae (L Dewaele, pers. obs., 2018; C de Muizon, pers. comm., 2018).

In anterior view, the proximal portion of the deltopectoral crest of IRSNB M2308 has a pronounced mammillary tuberosity, anteroproximal on the deltopectoral crest. Among Monachinae, this condition varies between a relatively smoothly-curving margin in Leptonychotes weddellii and a strongly pronounced mammillary tuberosity in Ommatophoca rossii. However, other known extant and extinct Monachinae show intermediate conditions, comparable to the condition in IRSNB M2308. Among Phocinae, this condition varies as well. Most Phocinae have to a certain degree a mammillary tubercle on the anteroproximal portion of the deltopectoral crest, with the exception of the hooded seal Cystophora cristata (Erxleben, 1777) and Pagophilus groenlandicus. This tubercle is strongly turned medially and can be assumed to be the insertion area of the atlantoscapularis muscle (see Howell, 1929; De Muizon, 1981). In lateral view, the trochideltoid surface is formed by the deltopectoral crest anteroproximally and the tricipital line (Evans & Lahunta, 2013) posterodistally. It extends from the greater tubercle proximally to the deltoid tubercle distally (De Muizon, 1981; Muizon & Bond, 1982). On IRSNB M2308 this surface is smooth and elongate, approximately twice as long as it is wide (approximately 6 cm long and 3 cm wide); its proximal and distal edges are rounded.

In posterior view, the diaphysis of IRSNB M2308 is roughly comparable in shape to most other Monachinae. On the posterior surface of the diaphysis, just distal to the humeral head and lesser tubercle, there is a moderately well-developed fossa for the origin of the medial head of the triceps brachii muscles. Among Monachine, De Muizon (1981) only observed a similar condition in Piscophoca pacifica. De Muizon (1981) also observed this condition in Frisiphoca aberrata (previously known as Monotherium aberratum), but recent phylogenetic analyses suggest that F. aberrata is not a monachine but a phocine seal (Dewaele, Lambert & Louwye, 2018). The distal end of the diaphysis and the distal epiphysis are missing. Only the most proximal portion of the supinator crest is preserved. The preserved portion indicates that this crest was poorly developed but massive as generaly observed in Monachinae. This supinator crest is consistently less developed in Monachinae than in Phocinae, in which it is sharp and well developed. However, it is noteworthy that the phocine condition is absent in the enigmatic Frisiphoca aberrata (Van Beneden, 1876; De Muizon, 1981; Berta & Wyss, 1994; Dewaele, Lambert & Louwye, 2018). Among Monachinae, the extinct Homiphoca spp. appear to have a well-developed crest, but not to the same extent as in Phocinae.

Discussion

Identification

The shape of the deltopectoral crest and trochideltoid surface of the humerus IRSNB M2308 supports the identification of the specimen as a monachine seal. Previously, it has been suggested that the distinction between fossil Phocinae and fossil Monachinae in the shape of the deltopectoral crest is not as clear as between extant Phocinae (abrupt distal termination approximately halfway the diaphysis) and Monachinae (smooth distal termination near the distal epiphysis) (e.g., Dewaele, Lambert & Louwye, 2017). However, Monachinae, both extant and extinct, are characterized by a roughly angular outline of the deltopectoral crest in lateral view and an insertion area for the pectoralis muscle extending along the distal half of the humerus, while this is not the case for Phocinae (Fig. 5). Corresponding to a curvature of the anterior margin of the deltopectoral crest much stronger in lateral view than the curvature of the posterior portion of the diaphysis, the roughly angular outline of the deltopectoral crest of IRSNB M2308, as well as the location of the insertion area of the pectoralis muscle on the humerus, suggests that this specimen represents a monachine seal. However, it is radically different from the other Pliocene monachines from the North Atlantic and Mediterranean, Auroraphoca atlantica, Callophoca obscura, Homiphoca spp., and P. etrusca: the maximum breadth of the deltopectoral crest is located relatively proximally. A. atlantica, from the early Pliocene Yorktown Formation at Lee Creek Mine, Aurora, North Carolina, differs strongly from IRSNB M2308 in the particular shape of the deltopectoral crest, extending much more distal, and the strong development of the lesser tubercle. The early Pliocene C. obscura, representing the stratigraphically second youngest monachine from the Antwerp area, next to IRSNB M2308, is noticeably larger and has a more robust humeral diaphysis. In addition, sexual dimorphism has been suggested for C. obscura, based on the size difference between the larger (junior synonym) Mesotaria ambigua and the smaller C. obscura. The specimen illustrated in Fig. 3E already represents a smaller morph of C. obscura. Given the described morphological differences between C. obscura and IRSNB M2308, this precludes identification of IRSNB M2308 as a sexual dimorph of C. obscura. Homiphoca spp. has a less pronounced deltopectoral crest. Representing the only contemporaneous monachine taxon to IRSNB M2308 from the Northern Hemisphere, P. etrusca differs notably in having a humeral head that strongly overlaps the diaphysis posteriorly, as well as a less developed deltopectoral crest. Consequently, humerus IRSNB M2308 most likely represents a new monachine species, the first known monachine from the latest early to late Pliocene of the North Sea (3.71 to 2.58 Ma), and thus the latest occurrence of Monachinae from higher latitudes of the North Atlantic (Fig. 6). Humeri have historically often been used as type specimens of phocids (e.g., Koretsky, 2001; Koretsky & Ray, 2008; Dewaele, Lambert & Louwye, 2018). However, we are reluctant to diagnose a new taxon, despite the presence of multiple characteristics that distinguishes IRSNB M2308 from other monachine humeri. This decision follows the suggestion from Dewaele, Lambert & Louwye (2018) that a humerus should be completely preserved to be acceptable as a type specimen. Hence, a proper diagnosis awaits more complete skeletal remains to be discovered.

Figure 6 Geographic distribution of late Miocene to recent Monachinae in the North Atlantic realm (including Mediterranean Sea).

Localities of fossil Monachinae are indicated by a black dot. Auroraphoca atlantica and Callophoca obscura are known from the late Miocene and early Pliocene of Antwerp, Belgium (C. obscura), and the early Pliocene of Lee Creek Mine, North Carolina, USA (A. atlantica and C. obscura) (Koretsky & Ray, 2008; Dewaele et al., 2018); specimen IRSNB M2308, Monachinae indet., from the late Pliocene of Antwerp, Belgium (this study); P. etrusca from the late Pliocene of Tuscany, Italy (Berta et al., 2015); and Pliophoca cf. P. etrusca specimens (grouped with P. etrusca for this figure) from the late Pliocene of Montpellier, France, and Riera du Bonet, Spain (Berta et al., 2015). Geographic ranges of the extant Monachus monachus and the recently extinct Neomonachus tropicalis are indicated in blue, following data presented by Jefferson, Webber & Pitman (2008) for M. monachus and Timm, Salazar & Peterson (1997) for N. tropicalis.

It is worth noting that the morphology of IRSNB M2308 most strongly resembles the morphology of the humerus of Piscophoca pacifica from the late Miocene of Sud-Sacaco, Peru, despite the strong geographical (North Sea Basin versus Southeastern Pacific Ocean) and temporal differences (latest early to late Pliocene for IRSNB M2308 versus late Miocene to earliest Pliocene for P. pacifica).

Biogeography

In the North Atlantic realm, monachine seals went extinct before the Pleistocene, with the exception of the extant M. monachus in the Mediterranean Sea, along the western shore of North Africa, and as far north as the northern shores of Spain (Deméré, Berta & Adams, 2003; González, 2015). Today, the higher latitudes of the North Atlantic Ocean are exclusively occupied by phocine seals (e.g., King, 1964; Jefferson, Webber & Pitman, 2008). Although the exact triggers of the extinction of Monachinae around the Pliocene-Pleistocene boundary in higher northern latitudes are unknown, multiple potential driving factors can be identified to explain this extinction. Ray (1976) suggested that Pliocene North Atlantic lineages of Monachinae could not adapt to decreasing seawater temperatures related to the global, Pliocene to Pleistocene decline in temperatures (see Zachos, Dickens & Zeebe, 2008). However, the presence of Monachus remains at relatively northerly latitudes (González, 2015) and the adaptation of lobodontin monachines to life in the Antarctic questions this assumption. It may equally be possible that their regional extinction at relatively high latitudes in the North Atlantic may be related to sea level changes, changes in the oceanic currents, trophic changes, or other environmental changes. Although Ray (1976) places the disappearance of Monachinae at relatively high northern latitudes around the early and late Pliocene boundary. Nevertheless, our finding suggests that the entire disappearance of Monachinae from relatively high northern latitudes must have occurred during the late Pliocene. Similarly, late Pliocene–Pleistocene climatic changes impacting the distribution and diversity of other groups of marine mammals, both regionally and globally, have been accounted for in the literature (Boessenecker, 2013; Churchill, Boessenecker & Clementz, 2014; Poust & Boessenecker, 2017; Slater, Goldbogen & Pyenson, 2017; Tsai et al., 2017), including the pinniped faunal turnover in the southeastern Pacific (Valenzuela-Toro et al., 2013). Different hypotheses regarding the causes of diversity changes across the Plio-Pleistocene boundary have been invoked. Ray (1976) and Deméré, Berta & Adams (2003) argued that North Atlantic lineages of Pliocene Monachinae did not evolve the pagophilic traits associated with ice-breeding observed in phocines and lobodontins in response to Pleistocene glacioeustatic events. In the context of that hypothesis, ongoing climatic change will most likely profoundly affect the survival and distribution of North Atlantic and Arctic phocids; relying on ice for pupping and nursing, pagophilic species are greatly threatened, whereas more temperate species may potentially broaden their range in higher latitudes (e.g., Johnston et al., 2012; Stenson & Hammill, 2014). Although a study of the paleobiogeographic evolution of Monachinae in response to climatic change is beyond the scope of this paper, Churchill, Boessenecker & Clementz (2014) showed that global temperature changes during the late Neogene and Quaternary were important drivers for changes in otariid biogeography. For cetaceans Marx & Uhen (2010) and Bisconti (2003) argued the presence of a link between higher primary productivity during the Pliocene than during the Quaternary, and reduced interspecific competition pressure. Consequently, more ecological niches were available during the Pliocene than thereafter. This reasoning may or may not be extrapolated to the evolution of Monachinae from the North Atlantic. Unfortunately, the present study is limited to specimen IRSNB M2308. And given the tentative identification, we deem it inappropriate to draw conclusions that are too far reaching.

Conclusions

Specimen IRSNB M2308 was discovered by Georges Hasse during construction works at the Lefèvre dock in Antwerp, Belgium, in the early 1900s. A reassessment of the geographic and stratigraphic settings and the local molluscan assemblage indicates that specimen IRSNB M2308 originates from the upper Pliocene Lillo Formation. This is the first latest early to late Pliocene phocid described from the higher latitudes of the North Atlantic realm (north of the Mediterranean). The subtriangular shape of the deltopectoral crest supports an attribution of the monachine subfamily, and the overall morphology indicates that the specimen does not represent either previously described early Pliocene monachines from the North Atlantic (Auroraphoca atlantica, Callophoca obscura, or Homiphoca spp.) or contemporaneous P. etrusca from the late Pliocene of the Mediterranean. This findind further increases the diversity of Monachinae during the Pliocene (and more specifically the late Pliocene), prior to the final extinction of the clade in higher latitudes of the North Atlantic.

This publication is in partial fulfilment of the PhD research project of Leonard Dewaele, under the supervision of Olivier Lambert and Stephen Louwye. The authors wish to thank Sébastien Bruaux, Cécilia Cousin, Alexandre Drèze, and Annelise Folie from the IRSNB (Brussels, Belgium), Christian de Muizon from the Muséum national d’Histoire naturelle (Paris, France), and David J. Bohaska and Nicholas D. Pyenson from the Natural Museum of Natural History (Washington, D.C., USA) for providing access to the collections of the respective institutions. We want to thank Christian de Muizon also for helpful comments on earlier versions of this manuscript, which greatly improved the quality of this work. We thank the academic editor, Nicholas D. Pyenson, and reviewers Robert W. Boessenecker, Morgan Churchill, Mario Cozzuol, and Ana Valenzuela-Toro for reviewing earlier versions of this manuscript and for helpful comments that improved the quality of the research presented here.

Institutional Abbreviations

IRSNB Institut royal des Sciences naturelles de Belgique, Brussels, Belgium

MNHN Muséum national d’Histoire naturelle, Paris, France

MSNUP Museo di Storia naturale, Università di Pisa, Pisa, Italy

USNM National Museum of Natural History, Smithsonian Institution, Washington, D.C., USA.

Additional Information and Declarations

Competing Interests

Author Contributions

Data Availability

The authors declare there are no competing interests.

Leonard Dewaele conceived and designed the experiments, performed the experiments, analyzed the data, contributed reagents/materials/analysis tools, prepared figures and/or tables, authored or reviewed drafts of the paper, approved the final draft.

Olivier Lambert and Stephen Louwye conceived and designed the experiments, analyzed the data, prepared figures and/or tables, authored or reviewed drafts of the paper, approved the final draft.

The following information was supplied regarding data availability:

The specimen studied, specimen IRSNB M2308, is part of the paleontology collection housed at the Royal Belgian Institute of Natural Sciences, Brussels, Belgium.

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
