# Peer review of "A late surviving Pliocene seal from high latitudes of the North Atlantic realm: the latest monachine seal on the southern margin of the North Sea"

_PeerJ, doi:10.7717/peerj.5734_

## Round 0.1 · original submission · Major Revisions

This manuscript has now received three excellent (and public) reviews. While one of the reviewers (Boessenecker) recommends acceptance-as-is (based on previous reviews of the manuscript for another journal), the other two recommend substantive, but in my view, fair revisions. On the balance, there is no good split between major and minor revisions, and the comments converge on matters concerning taxonomic comparisons between phocine and monachine postcrania. As a segue, the authors also need to address Boessenecker's core concern about conflicting philosophies for naming fossil pinniped taxa with respect to the corresponding author's recent RSOS publication asserting a contradictory view. Cozzuol also raises the obvious implications for paleobiogeographic patterns in phocid evolution should the authors demur from a hard stance on the material's assignment to Monachinae (Valenzuela Toro also echoed this issue). The authors need to address and likely revise the manuscript to meet all of these concerns for further consideration, and a revised manuscript will require further review from some or all of these reviewers.

Some additional detailed comments:

- Please avoid single quotations around common, non-technical terms such as 'northern seals,' as it evokes a euphemistic tone that isn't especially constructive for the arguments in the paper. Simply state some clades are considered more boreal or austral, based on extant distributions, while their fossil record strongly indicates otherwise (or a more complicated past) -- that's what's interesting here, for the non-specialist.

- Please control for spelling of taxonomic names. This is annoying work, but it's important: Properiptychus Ameghino, 1897 is misspelled, for example, on p. 5

- I am not particularly impressed by "in prep." citations, as they are essentially meaningless to the outside reader, and serve only to defend an argument or assertion from direct scrutiny. Consider instead "unpublished data," or "unpublished observations" -- more intellectually honest.

- Track for spelling errors in the figures captions (Fig. 2, "lithology," not "litholog," and Table 1 "analog," not "analogous"). If geopolitical boundaries on a map are noted (Fig. 4), they need to be identified on the map itself (e.g., U.S.A., Virginia, Italy, Belgium); also Table 1 "Character" is probably not the right word -- use "Measurement" instead. Also consider spelling out the articles used with the anatomical terms -- there are no limits for PeerJ, so please go the added the distance.

·

Basic reporting

The manuscript report an alleged “late” Surviving Monachinae seal in the southern margin of the North Sea. The material is restricted to a single, partial, humerus, lacking the distal end.

I see some issues with this manuscript that I will point following here and additional notes are inserted in the manuscript pdf.

Age.
The age, in general terms, was adequately established, but from reading the text, it seems that an age between 3.71 and 2.72My, thus, late Early to Late Pliocene is more accurate than just Late Pliocene.

Identification of specimen.
All the comparisons were done with Monachinae species, in the assumption that there is no doubt that it belong to this subfamily. As themselves noted, some features are not common to most (if not all) Monachinae, and even have one feature ( “a moderately well-developed fossa for the origin of the triceps brachii muscles” p.16, l.251) is also present in a species considered now as a Phocinae.

As the authors stated, in essence, the identification relies on the morphology of the deltopectoral crest (“The shape of the deltopectoral crest in the humerus IRSNB M2308 allows to identify the specimen as a monachine seal” p.16,l.264-265) which indeed, shows a condition most common in living and extinct Monachinae.

However, as the same authors noted in a previous publication:
“... many extinct Phocinae, such as C. maeotica, K. benegasorum, Phoca vitulinoides, and S. sintsovi (Cozzuol, 2001; Koretsky, 2001; Leonard Dewaele, 2016, personal observation) show varying conditions that are intermediate between extant non-phocid pinnipeds (Odobenus Brisson, 1762 and Otariidae Gray, 1825), monachines and extant phocines (Fig. 3). From what is present of the distal portion of the humerus, the deltopectoral crest of Prophoca rousseaui (and Leptophoca proxima) appears to terminate near the distal end of the diaphysis, as in O. rosmarus, otariids, and monachines (except L. weddellii and L. carcinophaga) and a number of extinct phocines (e.g., C. maeotica, K. benegasorum, Phoca vitulinoides, and S. sintsovi).” (Dewalele et al 2017).

Consequently, the condition of the deltopectoral crest in this specimen is not enough to assign this specimen to the Monachinae. Also, the presence of a prominent supinator crest is also unusual between Monachinae and the specimen described here had this condition, that may be considered an apomorphy of Phocinae. Unfortunately, the absence of the distal end of the humerus makes impossible to check the presence of a entepicondylar foramen, another derived feature of Phocinae.

All those makes necessary a better comparison and I think essential to add Phocinae, especially those that exhibit this (plesiomorphic) condition to it.

Monachinae geography.
One of the relevance of this finding is the late surviving of the subfamily in north coast of Europe. In the map of Figure 4 the authors plot the present and past distribution of Monachinae in the North Hemisphere. In it they show that Monachus monachus was restricted to the south and east of Mediterranean and to the Atlantic coast of North Africa. However, as was shown by Gonzales (2015), M. monachus was present in pre-historic times at least up to the Cantabrian coast, northern Spain, based on paleo-archeological material. Despite this no invalidate the relevance of the finding presented by the authors here, it shows that Monachinae were present until quite recent in localities much closer to the one form which the new specimen comes than they identified.

Experimental design

Not applicable

Validity of the findings

Since the feature used to determine the systematic position of the specimen is questionable, the central argument of the manuscript may be challenged and it needs, at least, be better justified.

Additional comments

Note on taxonomy.
Despite the division of the family Phocidae in two subfamilies (Monachinae end Phocinae) is widely accepted, in the last years several papers by Koretsky and co-authors proposed a different view, with four subfamilies (namely Monachinae, Phocinae, Cystophorinae end Devinophocinae, Koretsky & Rahmat, 2013, 2015, between others). As clearly the authors do not adhere to this view, I will like to see at least a mention of the reason for not.

Minor changes suggestions.
I suggest adding a small location map for the finding.
The reason for restricting the age to Late Pliocene contrary to late Early to late Pliocene should be better elaborate.

References.
Dewaele, L., Lambert, O. & Louwye, S. 2017. On Prophoca and Leptophoca (Pinnipedia, Phocidae) from the Miocene of the North Atlantic realm: Redescription, phylogenetic affinities and paleobiogeographic implications. PeerJ 5(5)

González, L.M. 2015. Prehistoric and historic distributions of the critically endangered Mediterranean monk seal (Monachus monachus)in the eastern Atlantic. Marine Mammal Science, 31(3): 1168–1192

Koretsky, I. & Rahmant, S. 2015. A new species of the Subfamily Devinophocinae (Carnivora, Phocidae) from the Central Paratethys. Rivista Italiana di Palentologia e Stratigrafia, 121(1):31-47

Koretsky, I. & Rahmant, S. 2013. First record of fossil Cystophorinae (Carnivora, Phocidae): Middle Miocene Seals from the Northern Paratethys. Rivista Italiana di Palentologia e Stratigrafia, 119(3):325-350

·

Basic reporting

The manuscript is understandable and professional. The structure of the manuscript follows the standard of the Journal. The different sections of the manuscript are adequately developed, allowing a fluid reading and understanding of the text. However, some points of the study need to be clarified (see below).

Experimental design

The main research subject is clear and relevant, improving the general knowledge of the pinniped fauna from the Pliocene of the North Atlantic Ocean.

Validity of the findings

See my comments below.

Additional comments

The most important issue is the fact that the authors do not explore the morphological differences between the humeri of Monachinae and Phocinae and neither perform comparisons of IRSNB M2308 with extinct members of Phocinae. This is relevant because it has been described that some fossil Phocinae exhibit “Monachinae” features (and also the other way around, with some species of fossil monachine with phocine features). See some examples in Figure 3 in Ray (1976), Cozzuol (2001), Wyss (1988). How would these observations affect the main conclusion of this study? Which other traits of the humerus are useful for differential diagnosis between the subfamilies? I suggest performing an extensive qualitative and quantitative comparison including species of Monachinae and Phocinae, with the aim to clarify this issue. Similarly, Fig. 3 should be improved by adding additional comparisons with relevant fossil Phocinae and Monachinae. Finally, please provide a table with comparative measurements of the humeri of the species mentioned in the text (e.g. Pliophoca etrusca, Callophoca obscura, Homiphoca, etc).

I also question about how the authors rejected the possibility that the differences between the humerus of Callophoca obscura and IRSNB M2308 (i.e. the overall size of the bone, the height of the lesser and the greater tubercle relative to the head, development of the pectoral crest) are not because of sexual dimorphism. I suggest, to include some type of analysis addressing this matter. How morphological variation because sexual dimorphism could affect some of your conclusions?

The discussion develops adequately the major implications of the finding. For clarity, I suggest to organize it in subsections (identity, biogeographic implications or so). Also, see my comments above. Finally, I suggest performing a deeper discussion about the biogeographic and diversity implications of this finding. For instance, is it consistent with classic reviews such as Ray (1976), Deméré et al. (2003)?

Other minor comments:
- Line 57-59, you state that "The elephant seals of the genus Mirounga Gray, 1827 live in sub-Antarctic waters and along the western shores of South America, but also in the Northeast Pacific, from California to Alaska”. However, permanent colonies of SES are found in Puerto Madryn, Argentina, which is located on the southeastern coast of South America.
- In the line 78 are mentioned Mediterranean, Caribbean and Hawaiian monk seals belonging together to the genus Monachus. However, a recent study (Scheel et al., 2014) proposed that Hawaiian and the extinct tropical monk seal belong to the new genus Neomonachus. Correct the assignation or justify why you do not.
- Line 178 says (Fig. 3A-D, versus Fig. 3E), but I should say Fig. 3F.
- Line 179 mentions the Fig. 4F, but it should be 3F.
- In Figure 3, be consistent with the anatomical views and information presented in the text. Further, I suggest improving contrast/light levels of IRSNB M2308, especially in anterior and lateral view.
- I suggest modify Fig. 4 and include the ages of the taxa next to the name of their corresponding species to make easier the visualization of the biogeographic and temporal consequences of this new finding.

·

Basic reporting

The manuscript is well-written in clear English, short/concise, and has an appropriate reference list.

Experimental design

No issues are evident with the methods. I do wonder about the statement regarding the suitability of naming a new genus and species from a holotype humerus - the authors are quite cautious here, yet in another recent paper, Dewaele and others (Royal Society Open Science) erect two new genera and species with humeri as holotypes. This contrasts starkly with the attitude expressed in the current paper, and I wonder if this bears more explanation.

Validity of the findings

The identifications in the paper are sound.

Additional comments

I reviewed an earlier draft of this paper, which was in already excellent shape. The authors have implemented all of my earlier suggestions, and therefore I recommend acceptance practically as-is. I note one very minor issue: It should be written as "analog caliper" in the table 1 caption. (is 0.1 mm really possible with non-digital calipers?).

Kind regards,

Robert W. Boessenecker, Ph.D.
College of Charleston
Charleston, SC

---

## Round 0.2 · Minor Revisions

This revision to the manuscript is improved over the original version. As with Reviewer 1 (Cozzuol), I may disagree with logic of the taxonomic assignment, but I think that the study is fine by PeerJ’s standards and that it provides a solid evaluation of an important specimen — in other words, it ought to be published, so that the discussion can move onwards. Reviewer 4 (Churchill) identified a few points that should be addressed in the next revision, especially regarding the squiggles in Figure 5 — I also understand what the authors are trying to capture, but it simply is not adequate, verifiable or robust. I recommend either enhancing the repeatability of that those comparisons with more sophisticated analyses, or simply ditching it and identifying it as a point for future work.

Equally, I wish for the authors to address the minor comments herein:

Abstract, Background: I insist that the authors qualify their description of extant monachine distribution to include a parenthetical remark about their presence in the Caribbean and Mediterranean -- especially because they make a big deal out of their North Atlantic fossil record. I suggest something along the lines of "..eastern Pacific (with historical distributions for Monachus in the Caribbean and Mediterranean)." The shifted baseline gives an entirely different biogeographic signal for extant trait distribution. Yes, the details are elaborated in the Introduction. But make the Abstract pass muster with any mammalogist.

L. 376, edit: ...lobodontines...

Acknowledgements, L. 411, please spell out Stephen's last name.

Figure 3 caption, use lithological instead of litholog

Figure 4, image, the term, in quotes, mamillary tuberosity, requires some explication. Please do so, or eliminate it from the figure.

Figure 6. Recommend italicizing ocean basin names, as per USGS guidelines -- they are graphically useful heuristics. Also, color is free with PeerJ: use it for the geographic distribution of extant Monachinae because N. tropicalis text is entirely obscured currently.

·

Basic reporting

The reply for authors are adequate enough despite I am still not completely convinced of the subfamiliar assignation of the specimen.
Despite this, I have no regrets in accept authors opinion and recommend the acceptance of the manuscript.

Experimental design

None

Validity of the findings

Acceptable.

Additional comments

None

·

Basic reporting

This paper is a straight forward study highlighting the latest known record of the Monachinae in the North Atlantic, providing further information on the biogeographic history of the group. Geological background and comparisons seem appropriate, and I have no reason to doubt the authors conclusions. References are mostly up to date, although I note that Figure 6 does not include references to Pliocene monachine records from Spain/Montepellier region of France, referenced in Berta et al. 2015 paper on Pliophoca. Figures seem good and English is also good, although I have a few minor grammatical suggestions I will list later.

Experimental design

Methods seem sound and the paper overall seems to fit the scope of the journal.

My only suggestion (which is likely outside the scope of these revisions, but something I strongly encourage the authors to pursue in the future), is that I would have like to seem some sort of quantification of the shape differences highlighted in Figure 5.For instance, generating an area measurement for the region under the curve representing the deltopectoral crest would allow better characterization of this variation and allow more rigorous comparison between taxa. Again, I only include this in here as something for the authors to pursue in the future, not something I expect them to perform for this revision.

Validity of the findings

Overall everything is sound, and most importantly I believe the referral of the specimen as far as taxonomic identity and age is correct.

I do think, in lines 359-363, the authors makes some pretty big assumptions that are a bit to speculative.I don’t think we can necessary assume it that sea temperature changes were the most important factor driving monachine extinction in the high latitude North Atlantic region, especially when the fossil record for the Late Pliocene and Early Pleistocene isnt that great for the region. Their extinction might be related to change in sea temperature, but given the success of the Monachinae in Antarctica, this seem to be a pretty big assumption, and other factors such as sea level changes may have played as large or larger a role, and maybe should be referenced.

Additional comments

The following are minor revisions, largely related to grammar or other small details:

Abstract, Results: “Confronting the information” = “Combining the information” Also in the abstract you mention the deltopectoral crest and other features as indicating monachine affinities. What are the other features?

Line 47: remove “commonly”

Line 52. Perhaps rephrase selected morphological features. After all, almost all morphologic classification is based on some set of features. Perhaps it would be more useful to use “few morphologic characters” instead.

Line 60: It would be useful to list what morphologic features clearly separate the group, rather than just state they are clearly separated.

Line 92: I thought the referral of fossils in the North Atlantic to Homiphoca has largely been refuted?

Line 187: “allowing to assume” = “allowing us to assume”

Line 219: “banded seal” = “harp seal”. I have never seen banded seal used as a common name in English.

Line 335: delete “the it should be noted that” as unnecessary

Line 376: replace “pagophilic evolution” with “pagophilic traits associated with ice-breeding”

---

## Round 0.3 · accepted · Accept

The authors have made a good effort to address the minor concerns by the reviewers. The resulting manuscript is improved and worthy of publication in PeerJ.

I have attached minor corrections for grammar and spelling, in PDF to this email - these can be addressed while in production.